

# Clinicopathological findings and imaging features of intraductal papillary neoplasms in bile ducts

Honghui Zhang[1,2,3,*], Zhendong Zhong[1,2,4,*], Gaoyin Kong[5], Junaid Khan[1,2,3], Lianhong Zou[6], Yu Jiang[6], Xiehong Liu[6], Yixun Tang[5], Bo Jiang[1,2,3], Chuang Peng[1,2,3], Yinghui Song[1,2,3] and Sulai Liu[1,2,3]

[1] Department of Hepatobiliary Surgery/Hunan Research Center of Biliary Disease, Hunan Provincial People's Hospital/The First Affiliated Hospital of Hunan Normal University, Changsha, Hunan province, China

[2] Biliary Disease Research Laboratory of Hunan Provincial People's Hospital, Key Laboratory of Hunan Normal University, Changsha, Hunan province, China

[3] Clinical Medical Technology Research Center of Hunan Provincial for Biliary Disease Prevention and Treatment, Changsha, Hunan province, China

[4] Department of Hepatobiliary Surgery, Changsha County People's Hospital/Hunan Provincial People's Hospital Xingsha Campus, Changsha, Hunan province, China

[5] Department of Anesthesiology, Hunan Provincial People's Hospital/Clinical Research Center for Anesthesiology of ERAS in Hunan Province, Changsha, Hunan province, China

[6] Key Laboratory of Study and Discovery of Small Targeted Molecules of Hunan Province, School of Medicine, Hunan Normal University/Hunan Provincial Institute of Emergency, Changsha, Hunan province, China

* These authors contributed equally to this work.

Corresponding authors
Yinghui Song, sissysyh@163.com
Sulai Liu, liusulai@hunnu.edu.cn

## ABSTRACT

**Background.** Intraductal papillary mucinous neoplasm of the bile duct (IPMN-B) is considered an uncommon tumor, and there is limited understanding of IPMN-B. This study aimed to investigate the prognosis and influential factors of the IPMN-B from 58 cases.

**Methods.** The clinical data of 58 patients with pathologically confirmed IPMN-B admitted to our hospital from January 1, 2012 to August 2017 were collected and analyzed. The patients were followed up by outpatient or telephone until January 1, 2019. SPSS 19.0 software was applied for data analysis. Survival analysis was performed using Kaplan-Meier method and parallel Log-rank test. Prognostic factors were analyzed by univariate analysis and multiple Cox regression model.

**Results.** Among of all the patients, 26 cases were benign tumors and 32 cases were malignant tumors. The preoperative tumor markers CA242 and CEA of malignant IPNM-B patients were significantly higher than those in benign tumors ($P < 0.05$). Survival analysis showed that patients with malignant tumors had a worse prognosis. The median survival time of malignant IPMN-B patients was $40.6 \pm 3.0$ months, yet median survival time of benign IPMN-B patients was not reached ($P = 0.19$). The one-year survival rate and three-year survival rate of benign IPMN-B were 84% and 74% respectively. The one-year survival rate and three-year survival rate of malignant IPMN-B were 88% and 64% respectively. Univariate analysis showed that combined lymph node metastasis, surgical method, and differentiation degree could affect patients' prognosis ($P < 0.05$). Multivariate analysis showed differentiation degree was an independent risk factor affecting prognosis (OR = 0.06, 95% confidence interval: 0.007~0.486, $P < 0.05$).
**Conclusion**. The levels of CEA and CA242 were helpful to identify benign and malignant of IPNM-B. Moreover, radical surgical resection could prolong patients' survival. Finally, differentiation degree was an independent risk factor affecting malignant IPNM-B prognosis.

## INTRODUCTION

Intraductal papillary mucinous neoplasm of the bile duct (IPMN-B) is considered an uncommon tumor, which secretes a large amount of mucus to cause significant expansion and obstruction of the bile ducts. *Kim et al. (2000)* described nine cases of bile duct tumors secreting a large amount of mucus and concluded that their clinical, imaging and pathological features were similar to those of papillary mucinous tumors in the pancreatic duct in 2000. In 2008, the concept of IPMN-B was first proposed to summarize such diseases (*Paik et al., 2008*). There is limited understanding of IPMN-B, which are mostly case reports, and lack of large-scale clinical research (*Nakanuma et al., 2016*; *Hokuto et al., 2017*). This study retrospectively analyzed 58 patients with IPMN-B admitted in our hospital from January 2012 to August 2017. Clinical and pathological data were collected and explored to identify the prognosis and influencing factors of IPMN-B in order to provide basis for the clinical diagnosis and treatment of IPMN-B.

## MATERIALS & METHODS

### Clinical manifestations

The clinical data of 58 patients with postoperative pathological diagnosis of IPMN-B were collected, including 24 males and 34 females; the median age was 61 years (41–85 years). The clinical manifestations were mainly pain in the right upper abdomen with jaundice. Abdominal ultrasound, CT, or MRI examinations were performed before surgery. Typical imaging findings were local or intrahepatic and extrahepatic bile duct dilatation, thickening of the bile duct wall with nodular tissue protrusions. Jelly-like bile was seen on duodenoscope examination in some patients. The study was approved by the Ethics Committee of Hunan Provincial People's Hospital/The First Affiliated Hospital of Hunan Normal University (Number: 2020–11), and all clinical samples were used in accordance with institutional guidelines and the Declaration of Helsinki after obtaining signed informed consent from all participants.

### Surgical methods

Preoperative auxiliary examinations were performed to evaluate important organ functions, liver reserve functions, and resectability of liver. Pathological histological examinations were performed during and after surgery. According to the results of pathological histological examinations and the degree of tumor invasion, they were divided into three subgroups: atypical hyperplasia, adenoma and adenocarcinoma (including cancerous changes). The

first two groups belonged to benign tumors. The surgical method was hepatic resection or combined tail resection according to the tumor location, and the intraoperative pathological results. The malignant tumors were routinely dissected with lymph nodes in groups 8, 12, 13a. Patients with multiple intrahepatic bile duct tumors by intraoperative choledochoscopy underwent hepatobiliary reconstruction, bile-gut drainage, and T-tube drainage. The specimens submitted including resected liver specimen, resection margins and lymph nodes.

### Follow up

The patients' gender, age, jaundice, biliary stones, previous biliary surgery, tumor markers (CA19-9, CA242, CEA), histological characteristics, lymph node metastasis, resection margins, etc. were followed up in outpatient and telephone visits after the operation. Recurrence was defined as both the recurrence of jaundice and new lesions on imaging. The end point of follow-up was the death of patients or the follow-up time until January 1, 2019, and those who died for other reasons but not IPMN-B were excluded.

### Statistical methods

SPSS19.0 software was applied for data analysis. Categorical data was compared using frequencies expressed as percentages, and compared with chi-squared testing. Tumor markers were analyzed by variance among subgroups. Overall survival analysis was performed using Kaplan–Meier method and parallel Log-rank test. Prognostic factors were analyzed by univariate analysis and multiple Cox regression model. $P < 0.05$ was considered statistically significant.

## RESULTS

### Clinicopathological parameters of IPNM-B patients

45 patients (77.6%) had bile duct stones, 35 patients (60.3%) had previous biliary surgery, and 38 patients (65.5%) had jaundice. Six of these 38 patients underwent percutaneous liver puncture of the biliary tract to remove jelly-like bile. The pathological results showed 14 cases (24.1%) were atypical hyperplasia, 12 cases (20.7%) were adenoma, and 32 cases (55.2%) were adenocarcinoma (including canceration). Lymph node metastasis did not occur in benign tumors, and lymph node metastasis occurred in 6 cases (18.8%) of malignant tumors. Radical resection was performed in 46 cases (79.3%), and palliative resection was performed in 12 cases (20.7%). Malignant tumor differentiation grade: 17 cases with high differentiation (53.1%), 9 cases with high-medium differentiation or middle differentiation (28.1%), 6 cases with medium-low differentiation or poor differentiation (18.8%). It was reported that CA19-9, CA242 and CEA were three recognized plasma tumor markers of cholangiocarcinoma (*Wongkham & Silsirivanit, 2012*). In this study, all of the three biomarkers in adenocarcinoma group were higher than the other two groups ($P < 0.05$). See Table 1 for details. Moreover, CA242 and CEA were more valuable in the diagnosis of cholangiocarcinoma (*Ni et al., 2005*). In this study, the area under the CA242 curve was 0.781, meanwhile, the area under the CEA curve was 0.728 ($P > 0.05$). It seemed

**Table 1  Clinicopathological parameters of 58 IPNM-B patients.**

| Clinical Features | Benign | | Malignant | | χ2 | P value |
|---|---|---|---|---|---|---|
| | N | % | N | % | | |
| Age (years) | | | | | 0.017 | 0.897 |
| ≤60 | 15 | 44.12 | 19 | 55.88 | | |
| >60 | 11 | 45.83 | 13 | 54.17 | | |
| Gender | | | | | 0.500 | 0.479 |
| Female | 17 | 48.57 | 18 | 51.43 | | |
| Male | 9 | 39.13 | 14 | 60.87 | | |
| Stones | | | | | 0.012 | 0.913 |
| Negative | 6 | 46.15 | 7 | 53.85 | | |
| Positive | 20 | 44.44 | 25 | 55.56 | | |
| Previous biliary surgery | | | | | 0.139 | 0.710 |
| Negative | 11 | 47.83 | 12 | 52.17 | | |
| Positive | 15 | 42.86 | 20 | 57.14 | | |
| Jaundice | | | | | 0.330 | 0.566 |
| Negative | 10 | 50.00 | 10 | 50.00 | | |
| Positive | 16 | 42.11 | 22 | 57.89 | | |
| CA19-9 | | | | | 0.051 | 0.821 |
| Normal | 13 | 44.83 | 16 | 55.17 | | |
| High | 13 | 41.94 | 18 | 58.06 | | |
| CA242 | | | | | 7.083 | 0.008 |
| Normal | 21 | 56.76 | 16 | 43.24 | | |
| High | 5 | 21.74 | 18 | 78.26 | | |
| CEA | | | | | 8.148 | 0.004 |
| Normal | 25 | 54.35 | 21 | 46.65 | | |
| High | 1 | 8.33 | 11 | 91.67 | | |
| Surgical method | | | | | 0.808 | 0.369 |
| radical resection | 22 | 47.83 | 24 | 52.17 | | |
| palliative resection | 4 | 33.33 | 8 | 66.67 | | |

**Notes.**

$\chi^2$ test was used to compare the distribution of clinical features between benign IPNM-B patients and malignant IPNM-B patients.

CA19-9 normal reference range: 0–35 U/ mL.

CA242 normal reference range: 0–20 U/mL.

CEA normal reference range: 0–5 ng/ mL.

A P value <0.05 was considered significant.

that CA242 was more sensitive and specific than CEA in diagnosing malignant IPMN-B from the ROC curve (Fig. 1).

All patients in this study underwent radical or palliative surgical treatment. Among them, 13 patients who underwent T-tube drainage and palliative intestinal drainage due to acute obstructive suppurative cholangitis (AOSC) or multiple tumors had a shorter survival time than those who underwent liver resection ($p < 0.05$). The median survival time of patients who achieved R0 resection was significantly longer than those of R1, R2 resection, meanwhile the recurrence rate of patients with R1, R2 resection was significantly

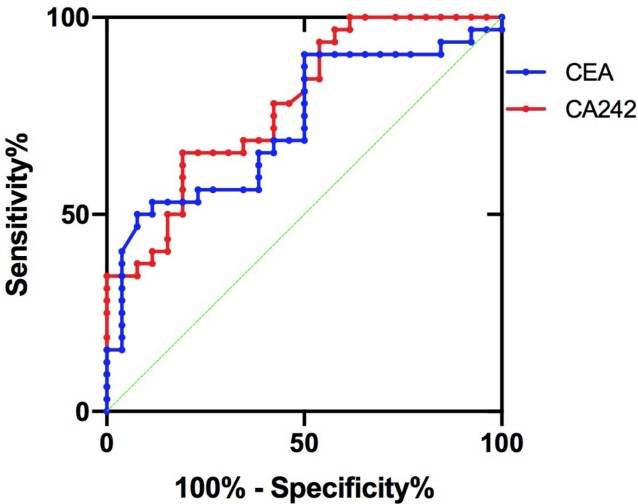

**Figure 1** ROC curve of CA242 and CEA.

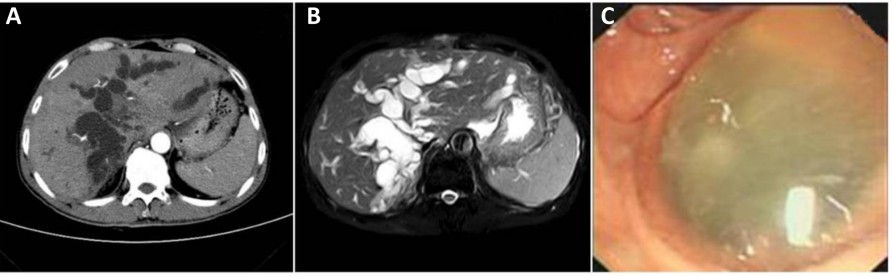

**Figure 2** Imaging of IPNM-B: (A) Typical CT manifestations showed obvious cystic dilatation of the intrahepatic bile duct, a small spot-shaped high-density lesion in the bile duct of the right hepatic lobe. (B) T2 magnetic resonance imaging showed a clear soft vine-like expansion of the intrahepatic bile duct, and a small nodular T2 signal foci in the bile duct. (C) The duodenoscopy showed a common bile duct duodenal fistula by below the duodenum above the duodenal papilla, about 1.0 cm in diameter, full filled with full of jelly-like Slime.

higher than that of R0 resection (R0 is complete microscopic resection, R1 is residual microscopic disease and R2 is residual macroscopic disease).

## Imaging of IPNM-B

In this study, massive jelly-like mucus could be seen in the bile duct drainage of most patients. Typical CT manifestations showed obvious cystic dilatation of the intrahepatic bile duct, a small spot-shaped high-density lesion in the bile duct of the right hepatic lobe (Fig. 2A). T2 magnetic resonance imaging showed a clear soft vine-like expansion of the intrahepatic bile duct, and a small nodular T2 signal foci in the bile duct (Fig. 2B). The duodenoscopy showed that a duodenal fistula with a diameter of 1.0 cm above the nipple which is filled with jelly-like slime (Fig. 2C).

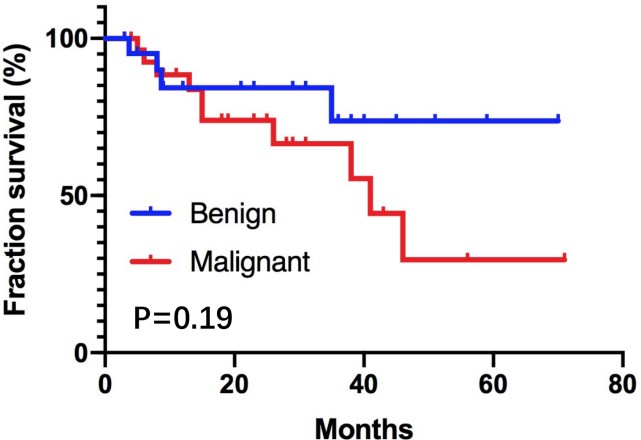

**Figure 3** The cumulative survival of patients with benign IPNM-B patients and malignant IPNM-B patients was determined by the Kaplan–Meier method.

## Survival analysis

Fifty-one cases (87.9%) were followed up to the endpoint, no perioperative deaths, the average follow-up time of these patients was $30.0 \pm 3.2$ months. Four patients with benign tumors died, of which two elderly patients survived 3.7 months and 8.7 months, respectively; two patients survived 7.5 months and 35.2 months due to multiple intrahepatic and extrahepatic bile ducts lesions, respectively. The overall mean survival time of this group was $56.4 \pm 6.0$ months. 10 patients with malignant tumors died, the median overall survival time was $41.4 \pm 6.1$ months. Kaplan–Meier survival curves of two groups were shown in Fig. 3. Log-rank test= 1.65, $P = 0.19$. The one-year survival rate and three-year survival rate of benign IPMN-B were 84% and 74% respectively. The one-year survival rate and three-year survival rate of malignant IPMN-B were 88% and 64% respectively. Univariate Cox proportional hazard regression analyses showed that lymph node metastasis, surgical method, and degree of differentiation were prognostic factors of malignant IPNM-B ($P < 0.05$). See Table 2 for details. Including lymph node metastasis, surgical method, and degree of differentiation into the COX model and performing multivariate analysis showed that differentiation degree (OR = 0.06, 95% confidence interval: 0.007–0.486, $P < 0.05$) was an independent factor affecting prognosis.

## DISCUSSION

IPMN-B is one of intraductal papillary neoplasm of the bile duct. It secretes a large amount of mucus to block the bile duct and causes obstructive jaundice. Histopathology of IPMN-B is similar to intraductal papillary mucinous neoplasm of the pancreas (IPMN-P) (*Nakanuma et al., 2017*; *Fukumura et al., 2017*). However, it is also found that IPNB arising in intrahepatic catheters is the biliary counterpart of IPMN, while IPNB arising in extrahepatic catheters is different from typical IPMN Yasuni (*Nakanuma, Kakuda & Uesaka, 2019*). Most patients experienced preoperative ultrasound examination showed significant bile duct dilatation, hypoechoic flocculent light masses in the bile duct, and

**Table 2  Multivariate analysis of factors contributing to overall survival in 29 malignant IPNM-B patients.**

| Variables | Univariate analysis | | Multivariate analysis | |
|---|---|---|---|---|
| | HR (95% CI) | *P* value | HR (95% CI) | *P* value |
| Age (<60 vs. ≥60) | 0.114 (0.002–6.305) | 0.413 | – | – |
| Gender (female vs. male) | 0.166 (0.012–2.216) | 0.838 | – | – |
| Stones (negative vs. positive) | 2.007 (0.166–12.283) | 0.540 | – | – |
| Jaundice (negative vs. positive) | 3.890 (0.138–25.550) | 0.074 | – | – |
| CA19-9 (≤35U /L vs. >35U/L) | 0.835 (0.061–11.335) | 0.099 | – | – |
| CA242 (≤20U /L vs. >20U/L) | 0.326 (0.063–1.678) | 0.116 | – | – |
| CEA (≤5ng/mL vs. >5 ng/mL) | 1.143 (0.086–9.181) | 0.097 | – | – |
| Lymph node metastasis (negative vs. positive) | 0.368 (0.054–2.519) | 0.015 | 0.442 (0.111–1.755) | 0.246 |
| Surgical method (radical resection vs. palliative resection) | 5.444 (0.337–27.848) | 0.018 | 0.081 (0.010–0.673) | 0.665 |
| Differentiation degree (high vs. medium-low) | 0.056 (0.007–0.451) | 0.000 | 0.06 (0.007–0.486) | 0.008 |

**Notes.**

Univariate and multivariate analysis of prognostic factors in 29 malignant IPNM-B patients included in the survival analysis.

Statistical analyses were performed by Cox proportional hazards regression. A *P* value <0.05 was considered significant. Italic indicates significant *P* values.

CI, confidence interval.

obvious space occupying. Typical CT or MRI imaging features were soft vine-like dilatation of the bile ducts inside and outside the liver, and multiple nodules in the bile duct wall.

Previous studies have suggested that CT play an important role in identifying benign and malignant diseases of IPMN-B (*Oki et al., 2011*; *Paik et al., 2008*). It is reported that "floating sign" in the bile duct of MRI imaging is a typical manifestation of IPMN-B (*Ying et al., 2015*). In general, the imaging characteristics of IPMN-B patients in this study were basically consistent with the reported imaging characteristics.

The tumor markers CA242 and CEA of most patients were significantly increased before surgery, and they were helpful to distinguish benign from malignant diseases. It seemed that CA242 was more sensitive and specific than CEA in diagnosing malignant IPMN-B in this study, however, this conclusion needed more samples to verify. It is more credible that combination all the three biomarkers to assess the benign and malignant disease.

IPMN-B often presents with multiple lesions. Intrahepatic bile duct is the most common site and accounts for about 84% of all the IPMN-B. When intrahepatic and extrahepatic bile duct tumors invaded to the hilar region often indicated an enhancement of tumor infiltration. Although the progress of bile duct and pancreatic papillary adenomas is relatively slow, the prognosis is often poor when multiple infiltrates appear (*Luvira et al., 2016*).

Radical surgical resection is considered the only way to cure bile duct malignancies. Even the addition of EGFR-mAbs to gemcitabine-based first-line chemotherapy dose not

significantly improve neither overall survival nor progression-free survival (*Rizzo et al., 2020*). All patients in this study underwent radical or palliative surgical treatment.

The original cells of IPNB are considered to be bile duct gland cells, which are distributed along the intrahepatic bile duct and extrahepatic bile ducts, showing a slow transition from adenoma to adenocarcinoma with fewer lymph nodes or distant metastases. The prognosis of IPMN-B is significantly better than any other types of cholangiocarcinoma (*Schlitter et al., 2014*; *Gordon-Weeks et al., 2016*). In this study, IPMN-B often accompanied by biliary stones and it progressed slowly, but once jaundice appeared, it was easy to induce AOSC. Therefore, it is necessary for patients with stones or previous history of biliary surgery to perform routine physical examination. Radical surgery could effectively improve the survival rate of patients with IPMN-B. The role of adjuvant treatment for IPMN-B is yet to be established (*Yeh et al., 2006*). Therefore, it is particularly essential to take the optimal surgical method. For multiple tumors involving the hilar area, in order to avoid the occurrence of biliary obstruction, bile-intestinal drainage may be necessary to increase the diameter of bile outflow channels.

In this study, there was different of the cumulative survival rate between benign IPMN-B patients and malignant IPMN-B patients. The median survival time of malignant IPMN-B patients was $40.6 \pm 3.0$ months, yet median survival time of benign IPMN-B patients was not reached ($P = 0.19$), the survival of patients in the malignant group was worse than that of the benign group, which was consistent with the previous studies of IPMN-B survival time (*Wang et al., 2015*; *Luvira et al., 2017*). But the difference between the two groups is not significant. It is considered that the number of cases in this study is small and the survival time of two older patients in the benign group is particularly short. In the future, we will continue to follow up patients with this disease in order to obtain more convincing data to gain a deeper understanding of IPMN.

## CONCLUSIONS

In short, preoperative CT and MRI is helpful to improve the detection rate of IPMN-B. Meanwhile, the levels of CEA and CA242 are helpful to identify benign and malignant of IPNM-B. Moreover, radical surgical resection could prolong patients' survival, when radical surgery is not available, unobstructed drainage is necessary. Finally, more data should be collected from more patients of IPMN-B and long-term survival follow-up to improve the diagnosis and treatment of IPMN-B.

### Funding

This work was financially supported by following funds: the Central Guidance of Local Science and Technology Development Fund (Grant No. 2018CT5008); the Project of Scientific Research of Traditional Chinese Medicine in Hunan (Grant No. 201809); The Hunan Provincial Natural Science Foundation of China (Grant No. 2019JJ50320/2018JJ3296); the Clinical Research Center for Anesthesiology of ERAS in

Hunan Province (No. 2018SK7001). The funders had no role in study design, data collection and analysis, decision to publish, or preparation of the manuscript.

## Grant Disclosures

The following grant information was disclosed by the authors:
Central Guidance of Local Science and Technology Development Fund: 2018CT5008.
Project of Scientific Research of Traditional Chinese Medicine in Hunan: 201809.
Hunan Provincial Natural Science Foundation of China: 2019JJ50320, 2018JJ3296.
Clinical Research Center for Anesthesiology of ERAS in Hunan Province: 2018SK7001.

## Competing Interests

The authors declare there are no competing interests.

## Author Contributions

- Honghui Zhang conceived and designed the experiments, prepared figures and/or tables, and approved the final draft.
- Zhendong Zhong conceived and designed the experiments, performed the experiments, prepared figures and/or tables, and approved the final draft.
- Gaoyin Kong and Junaid Khan performed the experiments, authored or reviewed drafts of the paper, and approved the final draft.
- Lianhong Zou and Xiehong Liu analyzed the data, prepared figures and/or tables, and approved the final draft.
- Yu Jiang analyzed the data, authored or reviewed drafts of the paper, and approved the final draft.
- Yixun Tang conceived and designed the experiments, performed the experiments, authored or reviewed drafts of the paper, and approved the final draft.
- Bo Jiang performed the experiments, prepared figures and/or tables, and approved the final draft.
- Chuang Peng analyzed the data, prepared figures and/or tables, authored or reviewed drafts of the paper, and approved the final draft.
- Yinghui Song and Sulai Liu conceived and designed the experiments, prepared figures and/or tables, authored or reviewed drafts of the paper, and approved the final draft.

## Human Ethics

The following information was supplied relating to ethical approvals (i.e., approving body and any reference numbers):

The study was approved by the Ethics Committee of Hunan Provincial Hospital/The First Affiliated Hospital of Hunan Normal University (Ethical Application Number: 2020-11).

## Data Availability

The raw measurements are available in the Supplemental Files.

## Supplemental Information

Supplemental information for this article can be found online at http://dx.doi.org/10.7717/peerj.10040#supplemental-information.

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
