# Peer review of "Clinicopathological findings and imaging features of intraductal papillary neoplasms in bile ducts"

_PeerJ, doi:10.7717/peerj.10040_

## Round 0.1 · original submission · Major Revisions

Please address all the reviewer comments but look at comments from reviewer 3 in particular detail.

·

Basic reporting

The review has a clear aim and is well written. There are a few grammatical concerns that I think may need some attention.

Line 69: mentions the word rattan and I the word may not be in the appropriate context as it seems the intent was to use it as an adjective.

Line 80: The word 'were' may need to be removed from the sentence "The first two groups were belonged to benign tumors."

Line 90: telephone visits may be an option instead of "telephone ways"


Lines 111-112: The following sentence needs to be reworded, as it does not seem to flow.
"In this study, patients showed typical massive jelly-like mucus by biliary drainage before surgery or intraoperative biliary tract."


Lines 116-118: Sentence needs to be reworded as it does not flow, needs attention to terms like 'electronic duodenum' and 'by below the' and 'full filled with full' and word slime should not start with capital S.

"The electronic duodenum showed a common bile duct duodenal fistula by below the duodenum above the duodenal papilla, about 1.0 cm in diameter, full filled with full of jelly-like Slime (Figure 1C)."


Line 125: the word 'were' should be removed from the sentence
"with malignant tumors were died"



Line 128: perhaps consider revealed the following factors

"revealed that prognostic factors"


Line 156: consider adding the word 'with' so that it reads as often presents with instead of "IPMN-B often presents multiple lesions."


Lines 157-158:
the word accumulate does not flow with the sentence, not sure if they meant to indicate involvement of bile ducts in the hilar region
"When bile duct tumors inside and outside the liver accumulate bile ducts in the hilar region often indicate an enhancement of tumor infiltration"


Line 169: consider changing word slowly to slow so that it reads as 'slow transition' instead of "slowly transition from adenoma"

Experimental design

The research question is well defined and is an interesting concept as it is relatively rare. The study design was a prospective one and the length of time as well as number of patients seemed adequate for a statistically significant yield. Methods used to perform statistical analysis were outlined well.

Validity of the findings

The article was well written with a clear aim and straightforward study design and the authors gave a concise discussion and conclusion as well. The methods that they used to perform their analysis show that their results were significant and valid. Conclusion recaptured their aim and tied the overall study findings together.

Additional comments

This is a good study and interesting to read. There are only a few minor grammatical concerns I had.

Reviewer 2 ·

Basic reporting

We read with interest this paper, which reports cases of intraductal papillary neoplasm of the bile ducts- It addresses a relevant topic, especially given the growing attention towards this form of rare biliary cancer.

The manuscript is quite well written and organized.
Figures and tables are comprehensive and clear. The introduction explains in a clear and coherent manner the background of this study. Despite this, English language could be improved, since some parts makes comprehension difficult. The manuscript would probably benefit from English language editing.

Experimental design

No comment

Validity of the findings

The results of the study are coherent with literature data. The discussion includes articles of historical importance in this setting. Therefore, we recommend to include results and references related the following papers:

- Rizzo A, Frega G, Ricci AD, Palloni A, Abbati F, De Lorenzo S, Deserti M, Tavolari S, Brandi G. Anti-EGFR Monoclonal Antibodies in Advanced Biliary Tract Cancer: A Systematic Review and Meta-analysis. In Vivo. 2020 March-April; 34(2): 479-488. doi: 10.21873/invivo.11798.

- Nakanuma Y, Kakuda Y, Uesaka K. Characterization of Intraductal Papillary Neoplasm of the Bile Duct with Respect to the Histopathologic Similarities to Pancreatic Intraductal Papillary Mucinous Neoplasm. Gut Liver. 2019 Nov 15;13(6):617-627. PMID: 30982236 . doi: 10.5009/gnl18476.

Additional comments

We believe this article is suitable for publication in the journal although minor revisions are needed.

The main strengths of this paper are that it addresses an interesting and timely question and provides a clear answer, with some limitations. Linguistic revision and the addition of other references are highly suggested

Reviewer 3 ·

Basic reporting

There are numerous basic English language errors:

In the first title page, the author name "su lai liu" appears twice is not capitalised.

In the second title page, the author name "Yu jiang" is not capitalised.

In the abstract, the word "with" should be removed after "a worse prognosis"

In line 51, "significant expansion and obstruction in clinical practice" is poorly written. I suggest "significant expansion and obstruction of the bile ducts".

in line 66 "yellow staining of the skin and sclera" should simply be 'jaundice'.

line 69 "soft rattan" is not a medical term. What does this mean?

line 70 "electronic imaging" is not a medical term. What does this mean? Is this meant to be endoscopy?

line 90 "Recurrence was defined as the recurrence of jaundice and imaging...confirmed recurrence" is incorrectly phrased. It should state "Recurrence was defined as both the recurrence of jaundice and new lesions on imaging."

line 116: "electronic duodenum" what does this mean?

line 120: "4 patients" should be "Four patients"

line 125: the word "were" should be removed

line 134: "and intrapancreatic a subtype)"- what do these words mean? They do not seem to make sense.

line 139: "B- ultrasound"- what exactly does this mean?

line 148: "were complained" should be "complained of"

line 148: what does "head biliary calculi" mean?

line 148: there should be no full stop after the word "calculi"


Several issues with structure:

The methods section contains a large amount of descriptive information and percentages that belongs in the results, not the methods (e.g. from line 70 to 74)

Line 161 to 167 belongs in the results, not in the discussion. Also, R0, R1, and R2 resection are never defined.

Line 137 to line 142 is repeating what has already been said in the results, and should thus be deleted (except for the information about ultrasound findings, which should be moved to the relevant paragraph in the results).

Experimental design

The statistical methods section is inadequate: it needs to state that categorical data was also compared using frequencies expressed as percentages, and compared with chi-squared testing.

The criteria used to differentiate the three subgroups of histological findings should be described. Was there a validated international system that was used?

The survival analysis methodology is flawed in line 121 onward. It is not stated if the survival curve is for disease free survival or overall survival. I assume it is disease-free survival, as two elderly patients with benign tumors died and were excluded from the survival curve. However in line 124 the authors state they are analysing "overall average survival time", in which case the elderly patients should not have been excluded (assuing they were excluded for non-cancer related deaths). Assuming then that the authors are actually analysing disease free survival, it is scientific convention to not exclude such deaths from survival Kaplan Meier curves, but censor them and mark the censored events as ticks or dashes on the curve.

Validity of the findings

Many unclear aspects to the results:

"MST of benign IPMN-B patients was not reached" is a confusing way to express the better survival of benign patients in the abstract and the end of the discussion. Their MST should be mathematically estimated to provide comparison to the malignant group. Alternatively, proportions of survival at a set time-point (e.g. the median follow-up time, or the one-year survival rate or three-year survival rate) can be compared via chi squared testing between groups.

Furthermore, in line 124, it is stated that regarding the benign group, "the overall average survival time of this group was 62.1 ± 5.1 months". How did the authors calculate this number, yet could not provide the MST in the rest of the manuscript? What do they mean by "average" survival time? Similarly, why do the authors provide the one-year survival rate and three-year survival rate of the malignant group, but not the benign from in this paragraph?

in line 155 "(date not shown)" is confusing. Do the authors mean "data not shown"? And if so, why have the authors chosen not to include this useful data? If thy have constructed a ROC curve why not include it?

Additional comments

There are too many basic errors in scientific writing, and incomplete results and statistical analysis, to render this publishable.

---

## Round 0.2 · Major Revisions

The sentence "R0 is a complete resection under a microscope, R1 is a residual under a microscope, and R2 is a residual under the naked eye" is poorly written. It should be rephrased "R0 is complete microscopic resection, R1 is residual microscopic disease and R2 is residual macroscopic disease".

Experimental design
The survival analysis methodology in line 134 onward remains highly flawed and the authors have provided a totally unacceptable answer as to why they decided to remove 2 elderly patients who died with benign tumours from the overall survival analysis.

The authors have replied to me that, when these patients were included, "the one-year survival rate of the benign group was 84%, meanwhile, the one-year survival rate of the malignant group was 88%, which was not in line with the objective development of the disease".

However, with these patients excluded, the one-year survival rate of the benign group was 94%, which I assume the authors feel is "in line" with the objective development of the disease.

But this is scientifically improper. Scientists and researchers cannot just deliberately exclude some of their collected data when they feel that these data do not fit a hypothesis! If the data do not fit a hypothesis, report this and explain it! (The obvious explanation here is that the study numbers are so small that 2 patients' deaths make a proportionally large impact.)

Validity of the findings
1) The information about this study's patients Lines 167 to 174 should be in the result section, not the discussion section.

The actual numerical data about CEA and CA242 levels between the two groups should be included in the manuscript, as well as the ROC curve that the authors presented to me in their rebuttal. After all, the abstract and conclusion mention CEA and CA242 extensively: this focus on CEA and CA242 needs to be adequately reflected in the manuscript.

2) where it is stated that "median survival time of benign IPMN-B patients was not reached" the authors should include the average follow-up time of these patients.

Reviewer 2 ·

Basic reporting

Nothing to add

Experimental design

Nothing to add

Validity of the findings

Nothing to add

Additional comments

Improved by recent changes we suggested.
We recommend acceptance of the article.
We congratulate the authors and we thank the Editor for the opportunity to revise this interesting paper.

Reviewer 3 ·

Basic reporting

All the previous English language errors I commented on have been corrected.

However, the new sentence "R0 is a complete resection under a microscope, R1 is a residual under a microscope, and R2 is a residual under the naked eye" is poorly written. It should be rephrased "R0 is complete microscopic resection, R1 is residual microscopic disease and R2 is residual macroscopic disease".

Experimental design

The survival analysis methodology in line 134 onward remains highly flawed and the authors have provided a totally unacceptable answer as to why they decided to remove 2 elderly patients who died with benign tumours from the overall survival analysis.

The authors have replied to me that, when these patients were included, "the one-year survival rate of the benign group was 84%, meanwhile, the one-year survival rate of the malignant group was 88%, which was not in line with the objective development of the disease".

However, with these patients excluded, the the one-year survival rate of the benign group was 94%, which I assume the authors feel is "in line" with the objective development of the disease.

But this is scientifically improper. Scientists and researchers cannot just deliberate exclude some of their collected data when they feel that these data do not fit a hypothesis! If the data do not fit a hypothesis, report this and explain it! (The obvious explanation here is that this study numbers are so small that 2 patients' deaths make a proportionally large impact.)

Validity of the findings

1) The information about this study's patients Lines 167 to 174 should be in the result section, not the discussion section.

The actual numerical data about CEA and CA242 levels between the two groups should be included in the manuscript, as well as the ROC curve that the authors presented to me in their rebuttal. After all, the abstract and conclusion mention CEA and CA242 extensively: this focus on CEA and CA242 needs to be adequately reflected in the manuscript.

2) where it is stated that "median survival time of benign IPMN-B patients was not reached" the authors should include the average follow-up time of these patients.

---

## Round 0.3 · accepted · Accept

Congratulation on the acceptance of your manuscript.

Reviewer 2 ·

Basic reporting

Nothing to add. The paper has been improved by recent changes.

Experimental design

Nothing to add.

Validity of the findings

Nothing to add.

Additional comments

Nothing to add. As I stated in the last revision I made, I believe the article should be accepted as is.

Reviewer 3 ·

Basic reporting

Now acceptable.

Experimental design

Now acceptable.

Validity of the findings

Now acceptable.